# One-Pot Synthesis of Double-Network PEG/Collagen Hydrogel for Enhanced Adipogenic Differentiation and Retrieval of Adipose-Derived Stem Cells

**DOI:** 10.3390/polym15071777

**Published:** 2023-04-03

**Authors:** Hwajung Lee, Hye Jin Hong, Sujeong Ahn, Dohyun Kim, Shin Hyuk Kang, Kanghee Cho, Won-Gun Koh

**Affiliations:** 1Department of Chemical and Biomolecular Engineering, Yonsei University, Seoul 03722, Republic of Korea; 2Departments of Plastic and Reconstructive Surgery, Chung-Ang University Hospital, Chung-Ang University College of Medicine, Seoul 06973, Republic of Korea

**Keywords:** adipose-derived stem cells, one-pot double-crosslinked hydrogel, cell-mediated degradation, adipogenesis, cell retrieval

## Abstract

Hydrogels are widely used in stem cell therapy due to their extensive tunability and resemblance to the extracellular matrix (ECM), which has a three-dimensional (3D) structure. These features enable various applications that enhance stem cell maintenance and function. However, fast and simple hydrogel fabrication methods are desirable for stem cells for efficient encapsulation and to reduce adverse effects on the cells. In this study, we present a one-pot double-crosslinked hydrogel consisting of polyethylene glycol (PEG) and collagen, which can be prepared without the multi-step sequential synthesis of each network, by using bio-orthogonal chemistry. To enhance the adipogenic differentiation efficiency of adipose-derived stem cells (ADSCs), we added degradable components within the hydrogel to regulate matrix stiffness through cell-mediated degradation. Bio-orthogonal reactions used for hydrogel gelation allow rapid gel formation for efficient cell encapsulation without toxic by-products. Furthermore, the hybrid network of synthetic (PEG) and natural (collagen) components demonstrated adequate mechanical strength and higher cell adhesiveness. Therefore, ADSCs grown within this hybrid hydrogel proliferated and functioned better than those grown in the single-crosslinked hydrogel. The degradable elements further improved adipogenesis in ADSCs with dynamic changes in modulus during culture and enabled the retrieval of differentiated cells for potential future applications.

## 1. Introduction

Adult stem cells possess self-renewal and pluripotent properties, making them ideal candidates for cell therapeutic applications aimed at facilitating tissue regeneration and repair. Adipose-derived stem cells (ADSCs) are particularly useful in research [1,2] due to their abundance and ease of acquisition through minimally invasive procedures [3]. ADSCs are highly pluripotent and can differentiate into multiple lineages, including adipogenic, osteogenic, chondrogenic, neural, hepatic, and myogenic lineages, making them a compelling choice for therapeutic applications [4,5].

While stem cells have great potential for cell therapy, their outcome can be further enhanced when combined with appropriate biomaterials, such as hydrogels. Rather than directly applying stem cells to tissue defects, hydrogels can be utilized as a method for cell delivery or as an environment for functional enhancement before their application [6]. In 3D culture, ADSCs have shown improved functionality and differentiation potential within a hydrogel environment [7]. However, contemporary research related to ADSC encapsulation has mostly focused on natural hydrogels such as alginate, gelatin, hyaluronic acid (HA), and heparin [8,9,10]. Although natural hydrogels possess excellent biocompatibility and biological resemblance to the extracellular matrix (ECM), they lack adaptiveness and exhibit limited mechanical properties compared to synthetic polymers [11,12,13]. Therefore, recent trends in 3D adipogenesis of ADSCs combine natural and synthetic components within a single hydrogel system, enhancing their desirable properties [14,15].

Another critical aspect to consider when utilizing a hydrogel system for ADSC engineering is the relaxation of hydrogel stiffness [16]. While the initial stiffness of the hydrogel is determined by the composition or concentration of polymers within the network at the time of gelation, the dynamic change in the mechanical properties of the hydrogel throughout the culture period is also critical in guiding the fate of encapsulated stem cells. Encapsulated stem cells can differentiate into various lineages depending on the stiffness of the hydrogels, as demonstrated by Chaudhuri et al. and Huang et al. [17,18]. Notably, adipogenesis favors a softer hydrogel, whereas osteogenesis is induced in stiffer hydrogels [19]. Therefore, as the hydrogel becomes softer, the encapsulated ADSCs can be effectively guided toward the adipogenic lineage. Such dynamic changes can be achieved through consistent degradation of the hydrogel network. Cell-mediated degradation is preferable, in particular, as it avoids physical disruption of the gel or the introduction of additional chemicals that might interfere with cell viability and functionality [20,21,22].

Here, we utilized bio-orthogonal chemistry to synthesize a double-network hydrogel consisting of polyethylene glycol (PEG) and collagen. This hydrogel was used to encapsulate and culture ADSCs, where cell-mediated degradation can lead to enhanced adipogenic differentiation of the encapsulated cells. The double-network hydrogel composed of PEG and collagen presented here possesses both suitable mechanical strength for cell confinement and reasonable cell adhesiveness, unlike a single-network hydrogel made solely of PEG or collagen. Additionally, the use of bio-orthogonal chemistries for gelation allows for the simultaneous formation of both PEG and collagen networks in a single pot, thereby avoiding time-consuming and labor-intensive multi-step procedures. In particular, the PEG gelation process, facilitated by a crosslinker containing disulfide bonds, allows for the tuning of the hydrogel properties, making it susceptible to cell-mediated degradation. ADSCs commonly secrete the antioxidant glutathione (GSH), which can rapidly untangle disulfide bonds through the thiol-disulfide exchange, leading to biodegradation [23,24]. This would consistently affect the stiffness of the hydrogel, which leads to enhanced adipogenesis and a simple and harmless method of cell retrieval using only collagenase. This system offers an improved culture system for adipogenic differentiation, as well as a safe method for retrieving functionally enhanced cells for future applications.

## 2. Materials and Methods

### 2.1. Materials

4-arm-PEG-succinimidyl glutarate (PEG-NHS, 10 K), phthalic acid buffer, 4-succinimidyloxycarbonyl-alpha-methyl-alpha (2-pyridyldithio) toluene (SMPT, 360.49 MW), 4-arm-PEG-NH_2_ (PEG-NH_2_, 10 K), 4-arm-PEG-SH (PEG-SH, 10 K), acetonitrile, Triton X-100, bovine serum albumin (BSA), Oil Red O, isopropanol (IPA), γ-L-glutamyl-l-cysteinyl-glycine (glutathione, GSH), collagenase, 2-[(2-hydroxy-1,1-bis (hydroxymethyl) ethyl) amino]ethanesulfonic acid (TES), calcium chloride, and 4-(2-hydroxyethyl)-1-piperazineethanesulfonic acid (HEPES) were purchased from Sigma Aldrich (Milwaukee, WI, USA). Phosphate-buffered saline (PBS, pH 7.4), Dulbecco’s phosphate-buffered saline (DPBS), and penicillin/streptomycin (P/S) were purchased from Gibco (Waltham, MA, USA). Slide-A-Lyzer^TM^ Dialysis Cassette (7000 Da MWCO), fetal bovine serum (FBS), and 4,6-diamidino-2-phenylindole dihydrochloride (DAPI) were purchased from Thermo Fisher Scientific (Waltham, MA, USA). Dimethyl sulfoxide (DMSO), sodium hydroxide (NaOH), and sodium chloride (NaCl) were purchased from Duksan Pure Chemicals (Korea). Pepsin soluble collagen (6 mg/mL) in 0.01 M HCl (6 mg/mL collagen solution) was purchased from Collagen Solutions (Eden Prairie, MN, USA). Trans-cyclooctene-PEG4-NHS (TCO-PEG-NHS) and methyltetrazine-PEG4-NHS (MTz-PEG-NHS) esters were purchased from Click Chemistry Tools (Scottsdale, AZ, USA). ADSCs were purchased from (American Type Culture Collection (ATCC), Manassas, VA, USA). Dulbecco’s modified Eagle medium with a low glucose concentration (DMEM-low) was purchased from Hyclone Laboratories, Inc. (Logan, UT, USA). Fluorescein-phalloidin (FITC-phalloidin), 3-(4,5-dimethylthiazol-2-yl)-2,5-diphenyl-2H-tetrazolium bromide (MTT), and the StemPro Adipogenesis Differentiation Kit were purchased from Invitrogen (Carlsbad, CA, USA). Paraformaldehyde (4%) was purchased from T&I Biotechnology (Seoul, Republic of Korea). The TaKaRa MiniBest Universal RNA Extraction Kit was purchased from Takara Bio, Inc. (Shiga, Japan). The ReverTra Ace^TM^ qPCR RT kit was purchased from Toyobo (Osaka, Japan).

### 2.2. Synthesis of Single-Crosslinked Hydrogels

#### 2.2.1. PEG (NHS) Hydrogel

PEG-NH_2_ (4.8 mM) was prepared in a solution of 0.3 M HEPES buffer and PBS mixed in a 1:1 ratio. PEG-NHS (6 mM) was prepared in a 10 mM phthalic acid buffer (pH 4.0) with 140 mM NaCl. The solutions of PEG-NH_2_ and PEG-NHS were then mixed in a 1:1 volume ratio and incubated at 37 °C for 20 min for complete gelation.

#### 2.2.2. PEG (SMPT) Hydrogel

PEG-NH_2_ was dissolved in PBS at a concentration of 25 mg/mL (2.5 mM), and SMPT was dissolved in acetonitrile at 40 mg/mL (0.11 M). After complete dissolution, the PEG-NH_2_ solution was mixed with the SMPT solution at a 2:1 volume ratio. The mixture was incubated at 25 °C for 2 h and dialyzed with Slide-A-Lyzer^TM^ Dialysis Cassette (7 kDa MWCO) overnight to remove unreacted monomers from PBS. The dialyzed mixture was then mixed with a 0.01 M PEG-SH solution in a 6:1 volume ratio and incubated at 37 °C for 20 min to produce a PEG (SMPT) hydrogel.

#### 2.2.3. Collagen Hydrogel

First, a neutralization solution was prepared by combining 1 N NaOH, distilled water, and 10× PBS at a 3:57:20 volume ratio. Then, 6 mg/mL collagen solution was neutralized by mixing it with the neutralization solution at a 17:5 volume ratio. The neutralized collagen solution was maintained below 4 °C until further use. A collagen-TCO (Col-TCO) solution was prepared by mixing 1 mL of the neutralized collagen solution with 22 μL of TCO-PEG-NHS ester dissolved in DMSO at 100 mg/mL. The collagen-MTz (Col-MTz) solution was prepared by mixing 1 mL of the neutralized collagen solution with 18 μL of MTz-PEG-NHS ester, which was also prepared as a 100 mg/mL solution in PBS. The collagen hydrogel was prepared by incubating a 1:1 volume ratio mixture of Col-TCO and Col-MTz solutions at 37 °C for 1 min.

### 2.3. One-Pot Synthesis of Double-Network Hydrogel

#### 2.3.1. PEG (NHS)-Collagen Hydrogel

The PEG (NHS)-Collagen hydrogel was synthesized by mixing PEG-NH_2_, PEG-NHS, Col-TCO, and Col-MTz solution at a 1:1:1:1 volume ratio. Each precursor solution was prepared, as described in Section 2.2. Gelation was completed by incubating the mixture at 37 °C for 20 min.

#### 2.3.2. PEG (SMPT)-Collagen Hydrogel

PEG (SMPT)-Collagen hydrogel was formed by mixing dialyzed PEG-NH_2_-SMPT (Section 2.2.2), PEG-SH, Col-TCO, and Col-MTz solution in a 6:1:3.5:3.5 volume ratio. The mixed solutions were incubated at 37 °C for 20 min to achieve complete gelation.

### 2.4. Rheological Characterization of Hydrogel

The rheological properties of the hydrogels were measured using a rheometer (MCR 302; Anton Paar, Ostfildern, Germany) using a constant shear strain of 1% with a frequency sweep from 0.1 to 10 Hz. To measure the storage modulus (G′) and loss modulus (G″) of the hydrogel under cell culture conditions, each hydrogel was incubated for 24 h and immersed in cell culture media under standard cell culture conditions (37 °C, 5% CO_2_, and 95% humidity).

### 2.5. Cell Culture and Encapsulation

ADSCs were cultured in DMEM-low containing 10% FBS and 1% P/S at 37 °C in a humidified 5% CO_2_ incubator. The medium was changed every two days. Cells at passage four were processed for encapsulation within the hydrogels. Before encapsulation, the hydrogel precursor solutions were sterilized under UV light for 1 h. The cells were detached from the culture dishes by trypsinization for 2 min and centrifuged at 1500 rpm for 5 min. Single cells were resuspended in hydrogel precursor solutions, and the mixture was dispersed in a 24-well plate to achieve a density of 50,000 cells/well. The hydrogel precursor containing the cell mixture was incubated at 37 °C for 20 min to complete gelation, and then 1 mL/well of culture media was added.

### 2.6. Analysis of Cell Morphology and Cell Viability

Cell morphology was observed at different time points (4 h, days 1, 3, and 5) after encapsulation using an optical microscope and fluorescence actin (F-actin) staining. For F-actin visualization, cell-encapsulated hydrogels were submerged in 4% paraformaldehyde for 15 min. The samples were washed thrice with DPBS for 5 min each, followed by permeabilization of 0.1% Triton X-100 for 1 h. After washing thrice, the samples were blocked with 1% BSA/PBS for 1 h. The samples were then incubated with a FITC-phalloidin solution for 1.5 h. Cell nuclei were counterstained with DAPI for 5 min at 25 °C. Stained cells were imaged using a confocal microscope (LSM 980; Carl Zeiss Inc., Thornwood, NY, USA). Phase-contrast images were obtained using an inverted microscope (Olympus, Tokyo, Japan).

Cell viability was quantified by the MTT assay after 4 h, and 1, 3, and 5 days of encapsulation. The culture medium was removed at the specified time points, and the cell-encapsulated hydrogel was washed twice with DPBS. MTT reagent was then added and incubated for 1 h. After the MTT reagent was removed, another DPBS wash was conducted twice, and DMSO was added. Purple formazan was dissolved in DMSO, and the absorbance of the reagent was measured at 540 nm using a single-mode microplate reader (SpectraMax ABS Plus; Molecular Devices, San Jose, CA, USA).

### 2.7. Oil Red O Staining

ADSCs encapsulated within the hydrogel were differentiated into adipocytes using the StemPro Adipogenesis Differentiation Kit for 14 days, according to the manufacturer’s protocol. After adipogenic differentiation, Oil Red O staining was performed to visualize the lipid deposition. The samples were then fixed in 4% paraformaldehyde for 15 min. After washing twice with DPBS and rinsing with 60% (*v*/*v*) IPA for 5 min, samples were stained with 0.3 *w*/*v*% Oil Red O solution (in 60 *v*/*v*% IPA) for 15 min. After staining, another rinse was conducted with 60% (*v*/*v*) IPA. The stained hydrogels were visualized under an inverted microscope.

### 2.8. QPCR Analysis

Hydrogels were homogenized manually with a pestle, and RNA was extracted using a commercial RNA extraction kit (TaKaRa MiniBest Universal RNA Extraction Kit), following the manufacturer’s protocol. cDNA was synthesized using a ReverTra Ace^®^ qPCR RT Kit. Gene expression was quantified by RT-PCR (CFX96; Bio-Rad Laboratories, Richmond, CA, USA) using the primer sequences listed in Table 1. The results were normalized to a housekeeping gene (β-actin) and analyzed using the ΔΔC_T_ method.

### 2.9. Cell Retrieval

GSH solution was prepared at 1 mM in PBS, and collagenase solution was prepared at 1 mg/mL in TESCA buffer (50 mM TES, 0.36 mM CaCl_2_, pH 7.4). The PEG (SMPT)-Collagen hydrogel was mixed with the prepared solutions in a 1:1 volume ratio, and the samples were stirred on a shaker for 5 min at 37 °C until complete degradation of the hydrogel was achieved. The degraded solution was centrifuged at 2000 rpm for 5 min to collect cells encapsulated within the hydrogel.

### 2.10. Statistical Analysis

All experiments were repeated at least thrice, and all data were expressed as mean ± standard deviation (SD). Statistical evaluation was performed by one-way analysis of variance (ANOVA) using GraphPad Prism (Version 8.4). Statistical significance was set at * *p* ≤ 0.05, ** *p* ≤ 0.01, and *** *p* ≤ 0.001.

## 3. Results and Discussion

### 3.1. Fabrication of Hydrogels

Five different types of hydrogels were fabricated for this experiment: Collagen, PEG (NHS), PEG (NHS)-Collagen, PEG (SMPT), and PEG (SMPT)-Collagen. In particular, the PEG (SMPT)-Collagen hydrogel was selected as the experimental group in this study because it has the most positive effect on ADSC adipogenesis.

As shown in Table 2, these hydrogels can be categorized based on the number of crosslinked networks (single or double) and degradability. ADSCs are encapsulated in each hydrogel type by mixing the individual hydrogel components with the cell suspension to produce a cell-encapsulated hydrogel structure.

For the non-degradable PEG (NHS) hydrogel (Figure 1a), 4-arm PEG functionalized with primary amine (NH_2_), or NHS was used to form the hydrogel network through crosslinking between NH_2_ and NHS [25,26,27,28]. Thiolated 4-arm PEG was used for the degradable PEG (SMPT) hydrogel instead of PEG-NHS to include SMPT as a crosslinker. The amine of PEG-NH_2_ binds to the NHS ester of SMPT, whereas the thiol group of PEG-SH binds to the pyridyldithiol part of SMPT through disulfide exchange, resulting in the final hydrogel network, where SMPT functions as a connecting link (Figure 1b) [23,29,30,31,32,33]. Because of the disulfide bond within the crosslinker, GSH secreted from ADSCs or added manually can break the bond and degrade the overall hydrogel structure. The collagen hydrogel was prepared by functionalizing collagen with TCO and MTz (Figure 1c). The click reaction between TCO and MTz is one of the fastest bio-orthogonal conjugative reactions, allowing collagen gel formation within a minute [34]. Double-network hydrogels consisting of PEG and collagen were synthesized simultaneously (one-pot) by processing the two different single-crosslinks mentioned earlier, for either PEG or collagen hydrogel (Figure 1d,e). Although hydrogel constructs were obtained using widely known chemical reactions with fully revealed mechanisms, further chemical analysis of different hydrogels with FTIR was conducted. The collagen hydrogel, PEG (NHS) hydrogel, PEG (SMPT) hydrogel, and hybrid hydrogels (PEG (NHS)-Collagen and PEG (SMPT)-Collagen) were distinguished from each other by their characteristic bands [35,36,37,38,39], while the two hybrid hydrogels had analogous IR spectra (Appendix A).

### 3.2. Rheological Properties of Hydrogels

The rheological properties of the fabricated hydrogels were analyzed using a rheometer to confirm gelation. The storage moduli (G′) of all five types of hydrogels were higher than their loss moduli (G″), indicating complete gelation (Figure 2a). The presence of the synthetic component (PEG) within the hydrogel significantly affected the storage modulus, as shown in Figure 2b. The collagen hydrogel exhibited significantly lower G′ values (8.0 ± 0.2 Pa) than other hydrogels. The non-degradable PEG (NHS) hydrogel showed a higher G′ value (96.8 ± 1.0 Pa) than the degradable PEG (SMPT) hydrogel (87.7 ± 0.6 Pa). Incorporating an additional collagen network into PEG-based hydrogel, which provides higher biocompatibility and cell-adhesive properties, resulted in a slight increase in storage modulus. Specifically, PEG (NHS)-Collagen had a storage modulus of 102.3 ± 1.0 Pa, and PEG (SMPT)-Collagen had a storage modulus of 98.3 ± 0.6 Pa. The mechanical properties of the hydrogels were significantly influenced by the presence of synthetic PEG hydrogels. PEG (NHS) and PEG (SMPT) hydrogels showed better mechanical properties than the natural polymer hydrogel composed solely of collagen. In contrast, the natural-synthetic hybrid hydrogels, PEG (NHS)-Collagen and PEG (SMPT)-Collagen, exhibited similar mechanical properties slightly higher than the synthetic PEG hydrogels. The changes in storage moduli according to the PEG were noted, and collagen ratios were also measured for both types of double-network hydrogels, namely, PEG (NHS)-Collagen and PEG (SMPT)-Collagen hydrogels. As shown in Figure 2c, the mechanical strength of the hydrogel increased as the ratio of PEG increased, confirming the contribution of PEG to the mechanical properties of the double-network hydrogel.

### 3.3. ADSC Culture within the Hydrogels

ADSCs were encapsulated within both single- and double-network hydrogels, and the proliferative capacity of the cells within the hydrogel was evaluated using the MTT assay (Figure 3a). The cells did not proliferate in the PEG (NHS) hydrogel, where the cell-adhesive motifs were limited. However, the cells showed substantial proliferation in both double-network hydrogels, likely due to the cell-adhesive property of collagen. Meanwhile, the PEG (SMPT) hydrogel was completely degraded within 24 h after encapsulation, and further measurements were not conducted beyond the first time point. As spontaneous degradation was expected to have occurred partially in the PEG (SMPT)-Collagen hydrogel due to GSH secretion from the ADSCs, the storage modulus of the PEG (SMPT)-Collagen hydrogel decreased to a greater extent than that of the non-degradable hydrogels (Figure 3b). The fluorescence images in Figure 3c show the different morphologies of ADSCs encapsulated within each type of hydrogel. The ADSCs cultured within the double-network hydrogel showed better proliferation and enhanced cellular morphology, demonstrating improved cell adhesiveness in the presence of collagen.

The pluripotent gene expression of ADSCs in the different hydrogels was observed, as shown in Figure 3d. On Day 1, there were no significant differences between the hydrogels. However, as the culture progressed, the expression of all three markers (Oct4, Sox2, and SSEA4) significantly decreased for both the PEG (NHS) and PEG (NHS)-Collagen hydrogels on Day 5 compared to those on Day 1. In the case of the PEG (SMPT)-Collagen hydrogel, a consistent increase in Oct4 and Sox2 expression was observed throughout the culture period, although the difference was not significant. The expression of SSEA4 remained constant from Day 1 to 5. However, on Day 5, the expression of all three markers was significantly higher in the PEG (SMPT)-Collagen hydrogel group than in the other two groups.

### 3.4. Adipogenic Differentiation in the Hydrogel

Adipogenesis was induced within the hydrogel for 14 days using a commercially available induction media kit. Oil Red O staining (Figure 4a), which indicates the presence of lipids in the cells by staining them red, revealed the highest efficiency of adipogenic differentiation within the PEG (SMPT)-Collagen hydrogel. PEG (NHS) and PEG (NHS)-Collagen hydrogels had a few lipids, but the extent was significantly lower than that in PEG (SMPT)-Collagen. ADSCs cultured within the PEG (SMPT)-Collagen hydrogel without active induction did not show any lipid formation (Figure 4a). This result was confirmed by qPCR (Figure 4b), where all three adipogenic markers (adiponectin, aP2, and PPAR-γ) were the highest in the PEG (SMPT)-Collagen hydrogels. The expression was approximately two-fold higher than that of the PEG (NHS)-Collagen hydrogel and ten-fold higher than that of the PEG (NHS) hydrogel.

### 3.5. Cell Retrieval

The images in Figure 5a demonstrate the complete degradation of PEG (SMPT), Collagen, and PEG (SMPT)-Collagen hydrogels upon the addition of GSH or/and collagenase. Figure 5b shows a decrease in the storage modulus of the collagen-containing hydrogel sample after treatment with collagenase, due to collagen degradation within the network. The storage modulus of SMPT-containing hydrogels could not be measured after treatment with GSH, as most hydrogels were heavily degraded beyond measurement. These results confirm that the initial mechanical properties of double-network hydrogels are primarily due to the PEG network.

The cytotoxicity of the cell retrieval procedures was confirmed by applying GSH and collagenase to PEG (SMPT)-Collagen hydrogels and analyzing the results using the MTT assay. (Figure 5c). The cells were treated with a 1 mM GSH solution, a 1 mg/mL collagenase solution, or a combination of both, in a 1:1 volume ratio for 5 min, followed by cell retrieval. Metabolic activity was measured 4 h, 1 day, and 3 days after the cell retrieval procedure. The cells exhibited reasonable viability and a similar proliferative capacity compared to the untreated control. The preservation of adipogenic characteristics in ADSCs cultured in PEG (SMPT)-Collagen hydrogel was further evaluated by qPCR analysis after cell retrieval (Figure 5d). Although no significant difference was observed in the expression of adiponectin and PPAR-γ, the expression of aP2 was higher when the cells were retrieved using GSH and collagenase treatment compared to the conventional method of mechanically disrupting the cells with a pestle.

## 4. Further Discussion

The PEG-Collagen hydrogel introduced in this study was synthesized using a one-pot method that utilized bio-orthogonal conjugative reactions to produce a double-network hydrogel that can be degraded through cell-mediated means. The use of an SMPT crosslinker containing a disulfide bond enabled the cell-mediated degradation of the gel through GSH secreted from encapsulated ADSCs. GSH is a tripeptide antioxidant consisting of glutamic acid, cysteine, and glycine. It is prevalent in the cytosol (up to 10 mM) and plays a pivotal role in antioxidant protection and metabolism. As the majority of glutathione within cells is in a reduced state, the thiol group in cysteine can break the disulfide bond through thiol-disulfide exchange via S_N_2 nucleophilic substitution [40,41]. As shown in Figure 6, the thiol from GSH attacks the sulfur of the disulfide bond within the SMPT crosslinker, forming a new disulfide bond [42]. Therefore, the PEG component within a single- or double-network PEG-Collagen hydrogel prepared with the SMPT crosslinker can be degraded upon ADSC encapsulation or GSH application.

ADSCs cultured within degradable double-network hydrogels showed significant proliferation while maintaining or enhancing pluripotency. The cleavage of disulfide bonds by secreted GSH caused the PEG components to lose their network, leading to a decrease in storage modulus (Figure 3b). Notably, the pluripotency of ADSCs in the PEG (SMPT)-Collagen hydrogel increased with decreasing mechanical strength (Figure 3d). This highlights the importance of degradability in maintaining and enhancing pluripotency in ADSCs. Partial degradation of the material facilitated cellular migration and provided more space for proliferation, as demonstrated by Schultz et al. [43,44]. Moreover, the breakdown of the PEG components allowed more collagen to be exposed to the encapsulated cells, increasing cell adhesion within the hydrogel. As stem cells maintain pluripotency through constant self-renewal, providing an environment where stem cells can readily proliferate is essential for pluripotency maintenance and improvement [45]. Therefore, ADSCs cultured within the degradable hydrogel exhibited enhanced pluripotency.

Adipogenesis of ADSCs was found to be further improved when cultured within a degradable hydrogel, as evidenced by the dynamic modulus of the hydrogel shown in Figure 3b. The decreasing modulus of the PEG (SMPT)-Collagen hydrogel throughout the culture promoted enhanced adipogenic differentiation of ADSCs. Many researchers have reported increased adipogenesis efficiency in relatively weaker hydrogels with a lower modulus, which is reasonable considering that adipose tissue is classified as soft tissue [19,46,47,48,49]. However, our study showed that the dynamic change in hydrogel modulus resulted in a superior environment for adipogenic differentiation, creating a stress-relaxation environment for the encapsulated ADSCs. As the degradation proceeds, the pressure exerted by the dense polymeric network on the encapsulated ADSCs begins to alleviate, allowing the cells to experience better flexibility in cell spreading, migration, and interaction with neighboring cells or the surrounding matrix [18,49]. This freedom of matrix mediation provided ADSCs with better adipogenesis within the PEG (SMPT)-Collagen hydrogel. Such phenomena resemble actual tissue regeneration that occurs in more dynamic and variable conditions in vivo.

For the investigation of adipogenic differentiation, three relevant marker genes, namely, Adiponectin, aP2, and PPAR-γ, were analyzed among various key adipogenic markers [50,51,52]. Adiponectin is the most abundantly secreted protein in adipose tissue [53], while aP2 is an adipocyte-specific fatty acid binding protein [54,55], and PPAR-γ is a protein related to the metabolic regulation of adipocytes [56,57]. PPAR-γ can be considered the main factor since it is the core regulator in adipocyte differentiation. If PPAR-γ is deficient, adipocytes cannot further differentiate into mature adipocytes, even when other strong pro-adipogenic factors are expressed [58].

In addition to the enhanced adipogenesis of encapsulated ADSCs, controlling the hydrogel degradation at the desired moment is critical for cell retrieval. After successfully differentiating stem cells towards the adipogenic lineage within the hydrogel, retrieving the cells is essential for further applications. Moreover, the complete degradation process of the hydrogel should maintain the viability of the cells and protect their functionality or differentiation potential. The encapsulated cells remained viable when GSH or collagenase was added to degrade the hydrogel (Figure 5c). Notably, the retrieved ADSCs exhibited further protection of their adipogenic characteristics compared to those cells retrieved through mechanical disruption (Figure 5d). This demonstrates the benefits of enzymatic or cell-mediated hydrogel degradation for cell retrieval prior to their application in cellular therapeutics.

## 5. Conclusions

In the present study, we successfully prepared a double-network PEG-collagen hydrogel in a single step for efficient adipogenic differentiation and retrieval of ADSCs. Synthetic PEG and natural collagen hydrogels complemented each other, with PEG providing mechanical support and collagen increasing bio-adhesiveness. Moreover, by tuning the polymer chemistry and crosslinking mechanism, we enabled cell-mediated degradation for the hydrogel’s regulatory stiffness, allowing complete retrieval of differentiated cells. As demonstrated in our study, hydrogel degradation guided by cell-secreted factors resulted in partial degradation of the hydrogel network in a manner consistent with cellular proliferation and maturation. The PEG (SMPT)-Collagen hydrogel, which had cell adhesiveness and cell-dependent biodegradability, exhibited the highest efficiency in cell culture and adipogenesis, as evidenced by both the cell proliferation and differentiation results. We anticipate that this one-pot, double-crosslinked PEG (SMPT)-Collagen hydrogel can enhance the cellular functions of other stem cell types and enable further applications of these cells through easy and safe retrieval procedures.

## Figures and Tables

**Figure 1 polymers-15-01777-f001:**
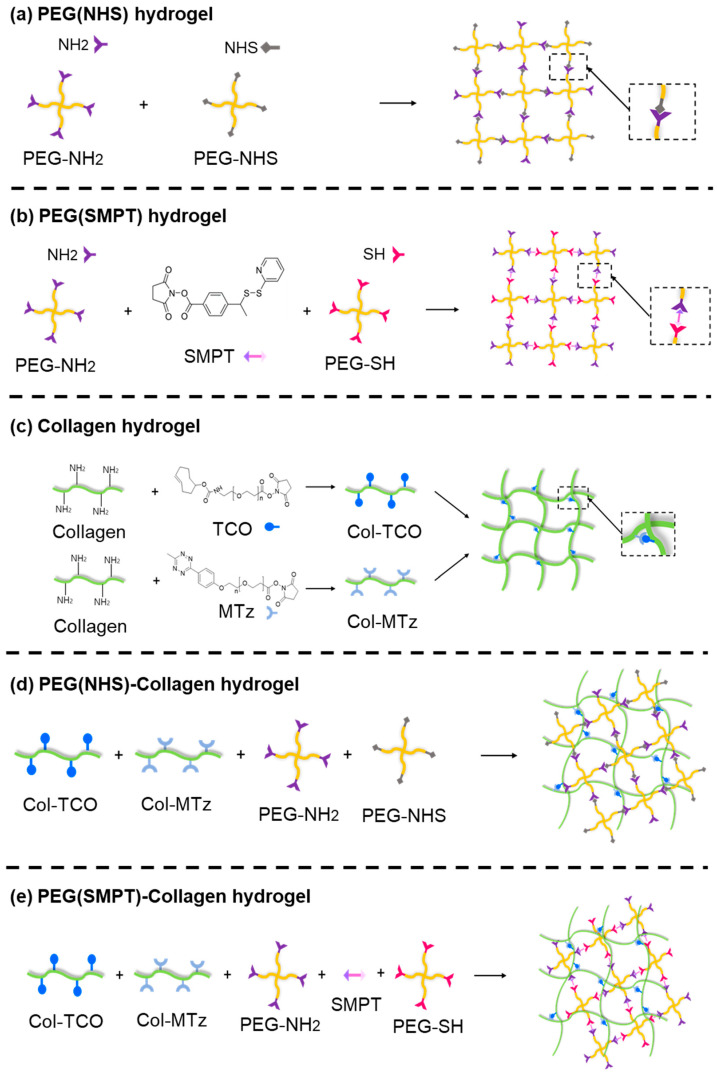
Schematic representation of chemical conjugation that constitutes each type of hydrogels: (**a**) PEG (NHS), (**b**) PEG (SMPT), (**c**) Collagen, (**d**) PEG (NHS)-Collagen, and (**e**) PEG (SMPT)-Collagen hydrogels.

**Figure 2 polymers-15-01777-f002:**
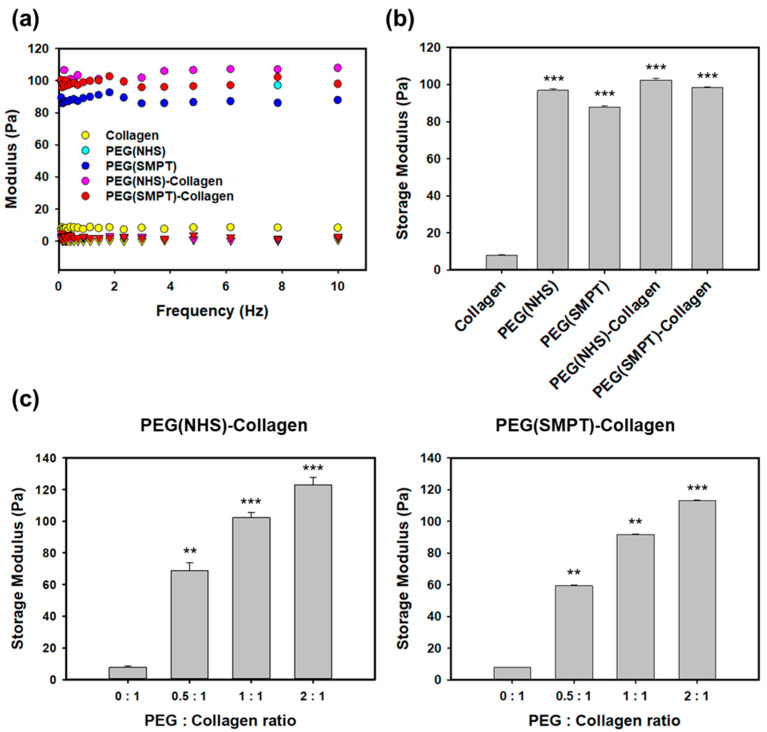
Rheological characterization of hydrogels. (**a**) Storage (●) and loss moduli (▼) of each hydrogel. (**b**) The averaged storage modulus of each hydrogel. *** *p* < 0.001 compared with Collagen. (**c**) Variations in storage modulus according to different PEG to collagen ratios for PEG (NHS)-Collagen (**left**) and PEG (SMPT)-Collagen hydrogels (**right**). ** *p* < 0.01, *** *p* < 0.001 compared with 0:1 PEG: Collagen ratio.

**Figure 3 polymers-15-01777-f003:**
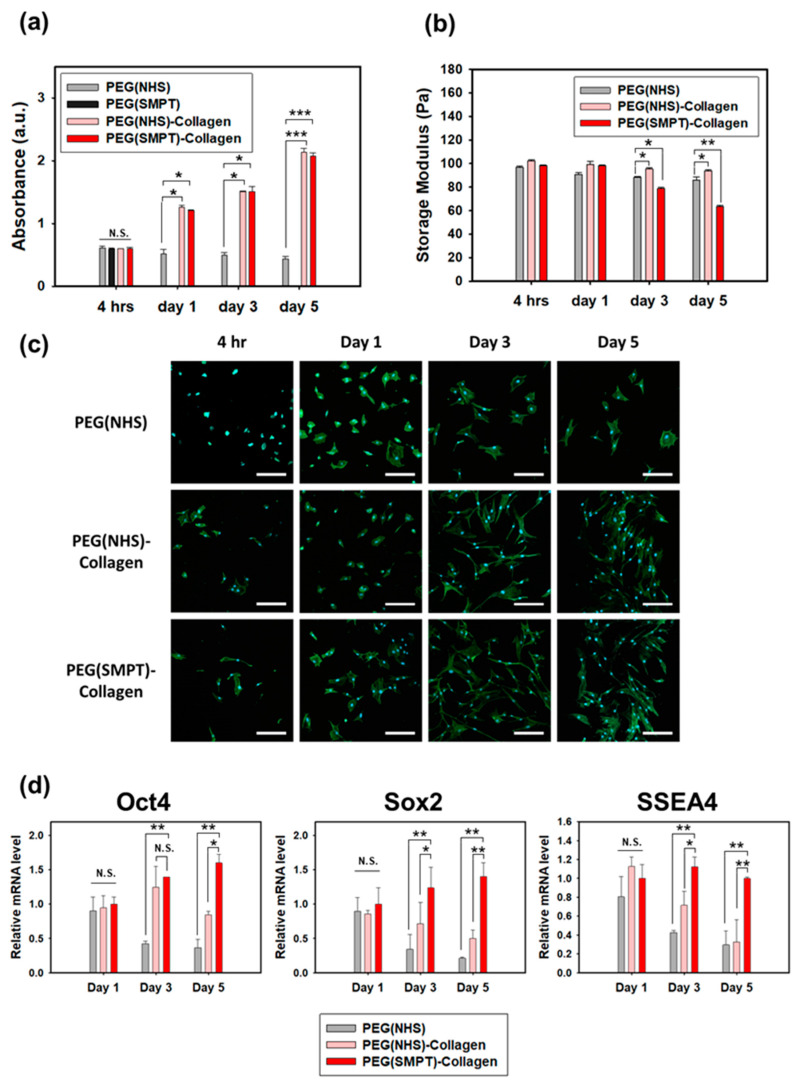
Characterization of ADSCs encapsulated within each hydrogel. (**a**) Proliferation of ADSCs in each hydrogel. * *p* < 0.05, *** *p* < 0.001 compared with PEG (NHS), and N.S. refers to non-significant. (**b**) Storage modulus of each hydrogel that was measured at different time points (4 h, 1, 3, and 5 days after the encapsulation). * *p* < 0.05, ** *p* < 0.01 compared with PEG (NHS). (**c**) Morphology of ADSCs encapsulated in each hydrogel observed by confocal microscopy after staining F-actin with FITC-labeled phalloidin and cell nuclei with DAPI. (**d**) The expression level of pluripotency markers (Oct4, Sox2, SSEA4) of ADSCs cultured in three different hydrogels, including PEG (NHS), PEG (NHS)-Collagen, and PEG (SMPT)-Collagen hydrogels. The gene expression levels were normalized to that of β-actin expression levels of the PEG (SMPT)-Collagen group on Day 1. * *p* < 0.05, ** *p* < 0.01 compared with PEG (SMPT)-Collagen, and N.S. refers to non-significant. Scale bar: 100 μm.

**Figure 4 polymers-15-01777-f004:**
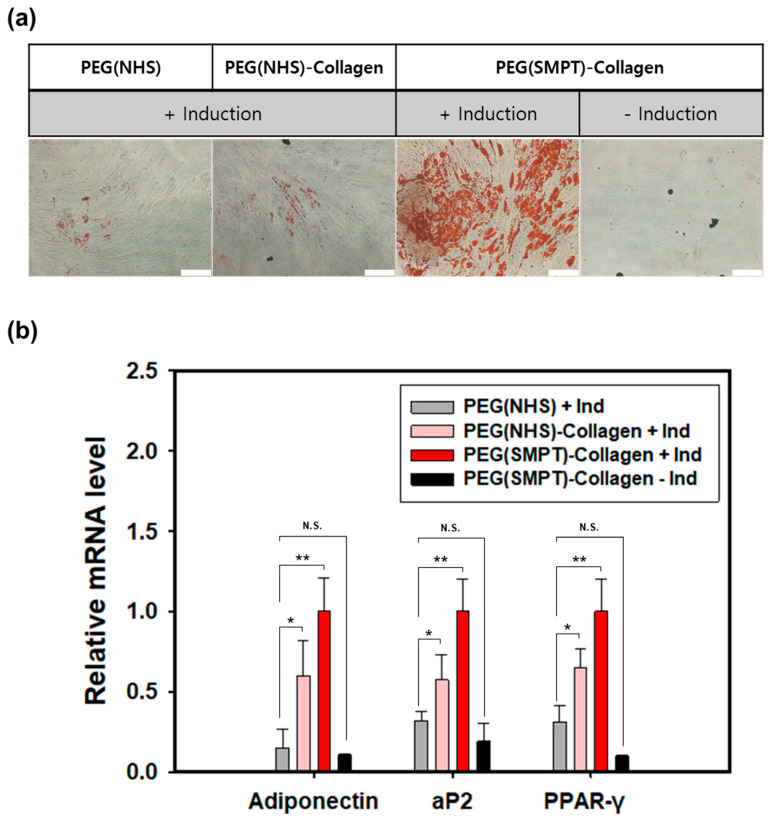
Adipogenic differentiation of ADSCs within each hydrogel. (**a**) Oil Red O staining of ADSCs within various hydrogels after adipogenic induction for 14 days. Scale bar: 200 μm. (**b**) The relative mRNA expression level of adipogenic markers (adiponectin, aP2, PPAR-γ) in 14 days of culture with induction (+Ind) media analyzed by RT-PCR. Gene expressions were normalized to that of β-actin and expression levels of the PEG (SMPT)-Collagen (+Ind) group. * *p* < 0.05, and ** *p* < 0.01 compared with PEG (NHS) with the induction media (+Ind), and N.S. refers to non-significant.

**Figure 5 polymers-15-01777-f005:**
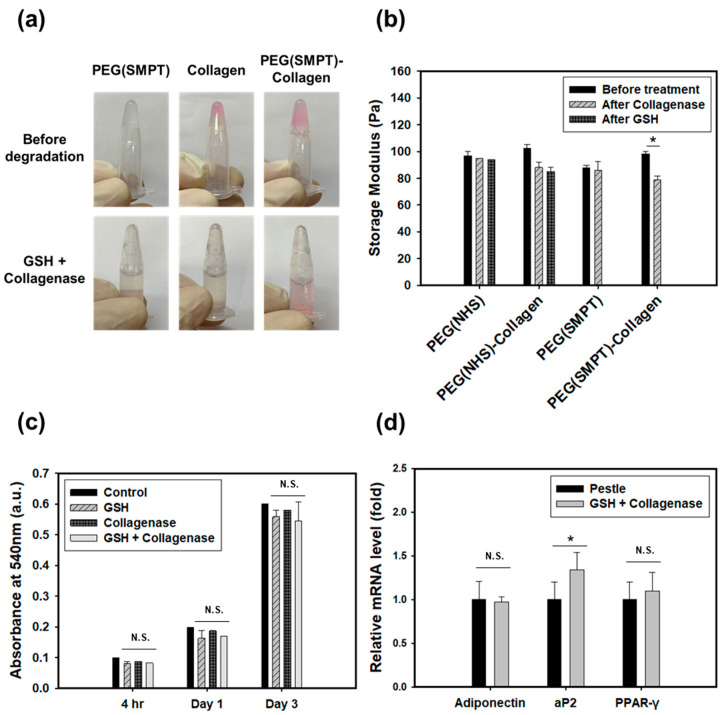
Cell retrieval by treating GSH and collagenase. (**a**) Photographic images of PEG (SMPT) hydrogel, collagen hydrogel, and PEG (SMPT)-Collagen hydrogel before and after simultaneous treatment of GSH and collagenase. (**b**) Storage moduli of various hydrogels before and after individually treating with collagenase and GSH. The initial collagenase treatment was followed by the subsequent GSH treatment. * *p* < 0.05. (**c**) The result of MTT assay at 4 h, 1 day, and 3 days after treating the cells with either GSH or collagenase or both. N.S. non-significant. (**d**) RT-PCR results comparing the adipogenic functions (adiponectin, aP2, PPAR-γ) between two methods of cell retrieval: pestle and GSH + collagenase treatment. * *p* < 0.05, and N.S.: non-significant.

**Figure 6 polymers-15-01777-f006:**
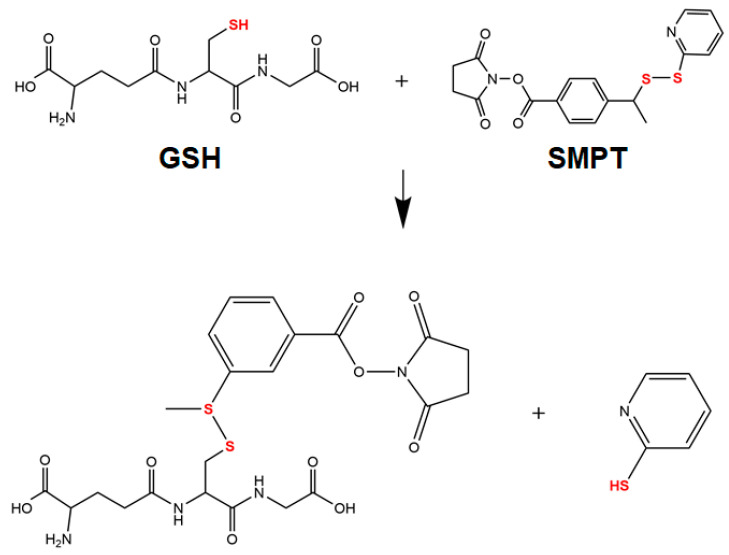
Representation of thiol-disulfide exchange reaction between glutathione (GSH) and SMPT crosslinker.

**Table 1 polymers-15-01777-t001:** Sequences of primers used in the qRT-PCR analysis.

Gene	Primer Sequences	
Β-actin	Sense	ACTACCTTCAACTCCATC
	Antisense	TGATCTTGATCTTCATTGTG
Oct4	Sense	ACATCAAAGCTCTGCAGAAA
	Antisense	CTGAATACCTTCCCAAATAGAAC
Sox2	Sense	TGCGAGCGCTGCACAT
	Antisense	GCAGCGTGTACTTATCCTTCTTCA
SSEA4	Sense	TGGACGGGCACAACTTCATC
	Antisense	GGGCAGGTTCTTGGCACTCT
Adiponectin	Sense	TGGTGAGAAGGGTGAGAA
	Antisense	AGATCTTGGTAAAGCGAATG
aP2	Sense	TGCAGCTTCCTTCTCACCTTGA
	Antisense	TCCTGGCCCAGTATGAAGGAAATC
PPAR-γ	Sense	ATGACAGCGACTTGGCAA
	Antisense	AATGTTGGCAGTGGCTCA

**Table 2 polymers-15-01777-t002:** Different hydrogel samples used in this study.

		Non-Degradable	Degradable
Single crosslink	Natural		Collagen
	Synthetic	PEG (NHS)	PEG (SMPT)
Double crosslink	Hybrid	PEG (NHS)-Collagen	PEG (SMPT)-Collagen

## Data Availability

The data presented in this study are available on request from the corresponding author.

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
