# Peer review of "One-Pot Synthesis of Double-Network PEG/Collagen Hydrogel for Enhanced Adipogenic Differentiation and Retrieval of Adipose-Derived Stem Cells"

_polymers, 2023, doi:10.3390/polym15071777_

Round 1

Reviewer 1 Report

The manuscript titled, “One-pot synthesis of double-network PEG/collagen hydrogel for enhanced adipogenic differentiation and retrieval of adipose-derived stem cells” the authors reported the one-pot PEG-collagen hydrogel formation using bio-orthogonal chemistry for effective adipogenic differentiation, cellular proliferation, maturation, and retrieval of adipose-derived stem cells using hydrogel degradation guided by cell-secreted factors. Therefore, the manuscript addresses the potential interest of the study.

However, the provided data sets to support the conclusion is not sufficient and thus it seems premature to proceed with the manuscript based on the current results. Hence, I recommend revision of manuscript to refine.

1.       The authors should add FTIR and NMR data for hydrogels.

2.       Electron microscopy images for the hydrogel should be added as it is essential for the characterization.

3.       qRT-PCR analysis for the expression of the key adipogenic marker genes such as PL1N, LPL, and FABP4 should be added.

Author Response

My co-authors and I would like to thank you and the reviewers for the thorough analysis of our manuscript. We mostly agree with your suggestions and revised the manuscript to address the comments made by the reviewers. Changes made in the revised manuscript were highlighted in yellow.

Reviewer 2 Report

The work entitled “One-pot synthesis of double-network PEG/collagen hydrogel for enhanced adipogenic differentiation and retrieval of adipose-derived stem cells” presents the development of a hybrid PEG-collagen hydrogel system that can be efficiently prepared via the one-pot synthesis strategy. The results are indeed relevant to the fields of tissue engineering and stem cell-based therapies, as they provide an alternative approach to fabricating highly bioactive scaffolds, by using fewer cytotoxic components and fabrication steps. Nonetheless, some issues remain to be addressed:

1.       The authors reported that hydrogel precursors were sterilized via UV exposure for 1 hour. Given that PEG and PEG derivatives are sensitive to light, how was precursor stability and integrity ensured after sterilization?

2.       Following the previous question, how would the authors explain the fact that the PEG(SMPT) hydrogel degraded completely in one day, whereas the PEG(SMPT)-collagen hydrogel remained in culture for 5 days?

3.       Lines 263-264: the authors stated that “…and the addition of collagen to PEG does not significantly alter the mechanical strength of the hydrogel”. Please clarify this, since based on the figure labels, what appeared to have been increased was the amount of PEG in the hydrogel. 

4.       Figures 5c and 5d do not specify which type of hydrogel the cells were retrieved from. Please clarify this.

5.       Figure 5d: the label for the vertical axis is not fully shown.

6.       Lines 74-76 and 79-82: please revise writing.

7.       About 48 % of the references in the manuscript are from 2013 or before. If possible, please include more updated references.

8.       There are some grammar mistakes throughout the manuscript. Please revise.

Author Response

(The authors gave the same response as above.)

Round 2

Reviewer 1 Report

The authors claimed that the specific hydrogel constructs that we introduced in this study are unprecedented and yet refused to provide the NMR data for the same. NMR data are essential requirement for any new compounds/biomaterials.

Furthermore, the authors refused to provide the electron microscopy data for the hydrogel which is basic characterization required before starting any experiments.

Moreover, the authors stated that PLIN, LPL, and FABP4 genes are key adipogenic markers that are abundant in adipocytes and yet refused to provide the data.

 I recommend rejection for this manuscript as basic characterization is missing from the manuscript.
